BEPCD: an ensemble learning-based intrusion detection framework for in-vehicle CAN bus

Xu Bocheng 18568967010@163.com
Cao Fei
Li Xilong
Tian Song
Deng Wenbo
Yue Shudan
Department of Cyberspace Security, Information Engineering University , Zhengzhou , China
Schiller Elad Michael
Electronic publication date: 2025 Aug 19
Publication date: 2025
Volume: 11
Electronic Location ID: e3108
Received 2024 Nov 18; Accepted 2025 Jul 16
Copyright: © 2025 Xu et al.
Copyright year: 2025
Copyright holder: Xu et al.
License: This is an open access article distributed under the terms of the Creative Commons Attribution License, which permits unrestricted use, distribution, reproduction and adaptation in any medium and for any purpose provided that it is properly attributed. For attribution, the original author(s), title, publication source (PeerJ Computer Science) and either DOI or URL of the article must be cited.
License URL: https://creativecommons.org/licenses/by/4.0/

Keywords: Intrusion detection system, Internet of vehicles, Controller area network, Ensemble learning, Vehicle security

Funding: National Key Research and Development Program of China 2021YFB3101804 Basic Strengthening Project of the Science and Technology Commission in the Field of Technology 2021-JCJQ-JJ-0496 This work was funded by the National Key Research and Development Program of China (2021YFB3101804) and the Basic Strengthening Project of the Science and Technology Commission in the Field of Technology (2021-JCJQ-JJ-0496). The funders had no role in study design, data collection and analysis, decision to publish, or preparation of the manuscript.

==============================
With the rapid development and widespread adoption of intelligent vehicles and the Internet of Vehicles (IoV), vehicle security has become a growing concern. Modern vehicles manage key components via the controller area network (CAN) connected electronic control units (ECUs). CAN bus intrusion techniques are the primary methods of compromising the IoV, posing a significant threat to the normal operation of critical vehicle systems, such as the power systems. However, existing attack detection methods still have shortcomings in terms of feature extraction and the diversity of attack type detection. To address these challenges, we propose an intrusion detection framework named basic ensemble and pioneer class decision (BEPCD). The framework first constructs a 15-dimensional feature model to hierarchically characterize CAN bus messages. Subsequently, BEPCD incorporates multi-model ensemble learning enhanced by a Pioneer class selector and confidence-driven voting mechanisms, enabling precise classification of both conventional and emerging attack patterns. Additionally, we analyze the importance of different data features across four machine learning algorithms. Experimental results on public datasets demonstrate that the proposed detection framework effectively detects intrusions in-vehicle CAN bus. Compared to other intrusion detection frameworks, our framework improves the overall F1-score by 1% to 5%. Notably, it achieves an approximately 77.5% performance enhancement in detecting replay attacks.

Introduction

Over the past decade, intelligent connected vehicle technology has advanced significantly (Zhang et al., 2023). The integration of multimodal data from diverse sensors allows vehicles to recognize their driving environment and make critical decisions. Core vehicle functions, such as acceleration, braking, steering, and engine management are primarily controlled by electronic control units (ECUs) (Wolf, Weimerskirch & Wollinger, 2007). The controller area network (CAN) is a widely adopted communication protocol for these essential ECUs, including those managing braking and throttle systems (Lokman, Othman & Abu-Bakar, 2019).

The CAN bus serves as the core transmission bus within the in-vehicle network (Jeong et al., 2022). However, due to the lack of authentication and encryption mechanisms in the CAN bus, it remains highly vulnerable to cyber-attacks (Lee et al., 2023). These attacks pose serious threats to the safety of the vehicle and its occupants. For instance, as early as 2008, Hoppe, Kiltz & Dittmann (2008) conducted experimental attacks on the in-vehicle CAN bus. In 2010, Koscher et al. (2010) were able to control the brakes and dashboard of a vehicle by attacking the in-vehicle CAN bus. Kulandaivel et al. (2021) exploited the peripheral clock gating feature of the microcontroller unit (MCU) to remotely manipulate the output of the ECU, thereby achieving an attack on the CAN bus. In 2014, Woo, Jo & Lee (2014) paired a smartphone with a Bluetooth OBD scanning tool to launch an attack on the CAN network and control the ECU, allowing the tampering of the dashboard, halting the engine, acceleration, and handle control functions. Miller (2015) successfully controlled the braking and steering of a Jeep Cherokee in 2015 through remote connection technology, leading to the recall of over a million vehicles with safety vulnerabilities by Chrysler. Additionally, Tencent’s Keen Security Lab (Nie et al., 2018) discovered multiple security vulnerabilities in Tesla models and infiltrated the driving control ECU via wireless connection. Given the potential of these attacks to endanger the life and property of drivers and passengers, the development of an IoV intrusion detection framework is of paramount importance (Ji et al., 2024).

Figure 1 shows the application scenarios of IoV intrusion detection systems. Machine learning methodologies have been extensively employed in the construction of in-vehicle CAN Bus intrusion detection frameworks. However, individual machine learning algorithms often demonstrate superior performance in identifying a specific type of attack (Yang et al., 2022; Aliwa et al., 2021). Therefore, ensemble learning has been widely used in the field of in-vehicle CAN bus attack detection. As early as 2019, Yang et al. (2019) implemented ensemble learning for in-vehicle CAN bus attack detection, attaining high detection accuracy while maintaining low computational overhead. Alalwany & Mahgoub (2024) employed a combination of stacking, bagging, and voting ensemble techniques to detect prevalent attacks on the in-vehicle CAN bus system. While ensemble learning-based intrusion detection frameworks have demonstrated promise, they suffer from critical limitations. First, existing approaches predominantly rely on global message frequency or payload entropy (Yang et al., 2019; Alalwany & Mahgoub, 2024), failing to model temporal dependencies (e.g., inter-message intervals) and localized contextual patterns (e.g., sequential ID relationships). This incomplete feature representation critically undermines replay attack detection (F1 <40% in LCCDE) due to their reliance on static payload signatures (Yang et al., 2022). Second, conventional ensemble strategies employ fixed-weight voting or stacking (Yeonseon et al., 2023), assuming uniform model performance across attack types. Such rigidity leads to suboptimal fusion of model-specific strengths and lacks decision transparency—opaque aggregation in frameworks like DivaCAN (Khan et al., 2024) and FFS-IDS (Altalbe, 2023) obscures model contributions, hindering trust in safety-critical automotive systems (Nwakanma et al., 2023).

Figure 1 The IDS-protected internet of vehicle.

In this study, we proposed to construct an efficient and robust in-vehicle CAN bus intrusion detection framework based on attack message characteristics. Specifically, we first extract features from CAN bus messages by leveraging their temporal regularity across different data types and validate their effectiveness. To further enhance the framework’s capability against diverse attack behaviors, our approach integrates machine learning models with a custom-designed decision-making algorithm. Finally, we quantitatively analyze the importance of individual data features across different machine learning models.

The main contributions of this research are as follows: Dynamic ensemble learning: Pioneer class selection and confidence-driven fusion: We innovatively designed an in-vehicle CAN bus intrusion detection framework. It combines Pioneer class methods, a confidence voting mechanism, and machine learning algorithms. Unlike traditional ones, the Pioneer class selector adaptively assigns model ensembles for different attack types by their feature profiles, and the confidence-driven voting mechanism refines classification by weighing predictions based on model confidence. This approach not only significantly enhances the adaptability of the framework but also remarkably improves its interpretability. It provides a clear decision-making process, facilitating a better understanding of how the framework classifies intrusion behaviors.

Developed a 15-dimensional feature model for in-vehicle CAN bus attack messages: We propose the 15-dimensional feature model specifically designed for in-vehicle CAN bus attack detection, addressing the limitations of fragmented feature engineering in prior works. By integrating temporal-contextual dependencies (preceding/succeeding message IDs, localized time intervals) and protocol-aware granularity (splitting payloads into eight dimensions), this model systematically captures stealthy anomalies in replay and spoofing attacks that evade conventional frequency-based detection. We employ DeepSHAP to quantify each feature’s contribution, enhancing interpretability and transparency for attack detection insights.

Comprehensive evaluation of the vehicular network intrusion detection framework: We evaluate the performance of the proposed intrusion detection framework using two public in-vehicle CAN bus datasets. Our framework achieves 1–5% higher overall F1-scores and 77.12% replay attack detection (77.5% improvement) via 15-dimensional feature model, validated by SHAP analysis. Rigorous feasibility evaluation confirms a lightweight architecture with 0.15 M parameters and 7.68 M MAC operations, and ensure that real-time requirements are met. Experimental results verify the accuracy and interpretability of our proposed framework.

The remainder of this article is organized as follows. “Related Work” reviews related work on vehicular network intrusion detection using ensemble models and feature extraction. “Preliminaries” covers the background and the context of the relevant research. “Proposed Framework” elaborates on the proposed BEPCD framework. “Experimental Work and Analysis” presents and analyzes the experimental results. “Discussion and Future Work” discusses the limitations of this study and outlines potential directions for future research. Finally, “Conclusions” provides a comprehensive summary of the research findings.

Related work

To address various attacks on vehicle networks, AI-based intrusion detection systems have become a mainstream solution. Researchers are particularly concentrating on developing integrated intrusion detection frameworks to boost the effectiveness of vehicle network intrusion detection systems.

Detection of various types of attack intrusions can be considered a multi-class classification problem. Bari, Yelamarthi & Ghafoor (2023) proposes an ML-based IDS for vehicle CAN buses, using Support Vector Machine (SVM), Decision Tree (DT), and K-Nearest Neighbor (KNN) classifiers on two real-world datasets to detect and classify various attacks with high accuracy. The study demonstrates the effectiveness of ML techniques in intrusion detection for vehicles, with DT showing superior performance, and highlights the importance of using multiple datasets to enhance the reliability and generalizability of the IDS. Park, Jo & Lee (2023) proposed Graph-based Intrusion Detection and Classification System (G-IDCS), a novel dual-stage framework for Controller Area Network (CAN) bus security. The system synergistically integrates a rule-based threshold detection module (TH classifier) with a machine learning-driven attack classification engine (ML classifier) through graph theoretical modeling. Derhab et al. (2022) presented H-IDFS, a histogram-based intrusion detection and filtering framework for in-vehicle networks. It uses traffic histograms to identify malicious windows and filters normal CAN packets from them via a novel one-class SVM, achieving 100% classification accuracy and filtering 94.93% to 100% of normal packets from malicious windows. Devnath (2023) was the first to apply the Graph Convolutional Network (GCN) to the intrusion detection of CAN bus data. By extracting features from CAN bus messages, the GCN can effectively detect mixed attacks. However, the study only used a single dataset, resulting in insufficient generalization ability of the model. Rai et al. (2025) used models like Long Short-Term Memory (LSTM), Gated Recurrent Unit (GRU), and Visual Geometry Group-16 (VGG-16) to learn the temporal features of CAN bus data for intrusion detection. Although their experiments showed that VGG-16 performed outstandingly in multi-class classification, achieving an accuracy of 100% in some cases, the types of attacks covered by their dataset were incomplete, and replay attacks were not included. Agbo (2024) proposed a hybrid model integrating D-CNN and Bi-LSTM, where the 1D-CNN captures spatial features and Bi-LSTM models temporal dependencies, achieving improved detection accuracy for complex attacks. However, the model exhibits high computational demands. Suwwan et al. (2021) employed 1DCNN, LSTM, and GRU networks, incorporating the time difference feature of data packets, achieving nearly perfect F1-scores on a specific dataset. However, this study has limitations. It relies solely on a single public dataset, casting doubt on the model’s generalization ability. Muzun, Javed & Rana (2024) proposed an innovative multi-stage intrusion detection system that combines a supervised artificial neural network (ANN) and an unsupervised long short-term memory autoencoder (LSTM-autoencoder). The system demonstrates excellent detection performance. However, it fails to detect replay attacks.

On the other hand, the emergence of explainable AI (XAI) is driven by the imperative to integrate model interpretability and decision traceability into autonomous systems, addressing safety-critical demands in IoV (Nwakanma et al., 2023). Seonghoon et al. (2023) proposed X-CANIDS, a signal-aware and explainable intrusion detection system. By parsing the payloads in CAN messages into understandable signals and integrating with a self-supervised autoencoder, it can effectively detect zero-day attacks. However, the system’s detection latency can be up to 73.2512 ms, which is relatively high compared to the minimum time interval (10 ms) of existing data streams in vehicles. Kayode & Chukwuere (2024) proposed a novel XAIEnsembleTL-IoV model architecture and utilized the Deep SHAP mechanism to provide a clear explanation of the model’s decision-making process. However, this work only used one dataset, which contained a limited number of attack types.

One important approach is using ensemble methods, which integrate multiple effective machine learning techniques into a multi-classifier system. Ensemble model for reliable intrusion detection system is a good choice for accuracy and explainability (Ahmed et al., 2024). For example, Khan et al. (2024) proposed a method called DivaCAN. DivaCAN employs several classifiers, including Multilayer Perceptron (MLP), gradient boosting machines, extra trees, random forests, and so on, to detect intrusion attacks on the in-vehicle CAN bus. DivaCAN shows good performance in detecting DoS and fuzzing attacks but performs poorly against zero-day attacks. Yang et al. (2022) introduced an ensemble method named LCCDE, which integrates three advanced machine learning algorithms XGBoost, LightGBM, and CatBoost and validated its effectiveness on two public in-vehicle CAN bus datasets. This ensemble method achieved near-perfect detection results but did not detect replay attacks. Alalwany & Mahgoub (2022) designed an effective IDS solution based on ensemble learning using eight supervised ML algorithms and an ensemble classifier. In experiments with in-vehicle CAN bus traffic datasets, this solution demonstrated high effectiveness in detecting both normal and abnormal activities. The authors also evaluated three fundamental ensemble learning methods and found that ensemble learning models indeed offer better performance by integrating multiple models compared to single models. Altalbe (2023) proposed the FFS-IDS system, which innovatively integrates multiple features extracted from raw network traffic and applies a stacking ensemble learning method to construct a more powerful classification model. However, this work only uses a single car-hacking dataset for model training and evaluation, so its generalizability remains questionable. Table 1 shows the comparative analysis of BEPCD and existing methodologies.

Table 1 A comparative analysis of BEPCD and existing methods across five key dimensions.

Method	Types of attack	Strategy	Explanation	Features	Limitation	
Bari, Yelamarthi & Ghafoor (2023)	3	ML	No	4	Static and single models	
Park, Jo & Lee (2023)	4	DL+ Rule based	No	5	Overlooks CAN message periodicity	
Derhab et al. (2022)	3	Histogram-based	No	3	The coverage of attack types is insufficient.	
Yang et al. (2019)	3	ML + Ensemble	No	9	Static ensemble strategy	
Khan et al. (2024)	4	ML + Ensemble	No	4	The coverage of attack types is insufficient.	
Yang et al. (2022)	3	ML + Ensemble	No	9	Static ensemble strategy	
Alalwany & Mahgoub (2024)	3	ML + Ensemble	No	4	Static ensemble strategy	
Yeonseon et al. (2023)	5	DL + Ensemble	No	13	Static ensemble strategy	
Rai et al. (2025)	4	DL	No	10	Model complexity	
Agbo (2024)	6	DL	No	10	Low accuracy	
Suwwan et al. (2021)	4	DL	No	NA	Single dataset	
Muzun, Javed & Rana (2024)	4	FL	No	9	Low generalization ability	
Seonghoon et al. (2023)	5	ML	Yes	107	High latency	
Kayode & Chukwuere (2024)	3	DL	Yes	NA	Fewer datasets	
Altalbe (2023)	3	DL + Ensemble	No	11	Static ensemble strategy	
Ours	4	ML + Adaptive decision making	Yes	15	–	

In summary, the construction of intrusion detection frameworks using Ensemble learning models has been demonstrated to significantly enhance detection efficiency for in-vehicle CAN bus systems. However, most existing frameworks inadequately address the unique characteristics of CAN bus attack messages. For instance, the extraction of features from raw CAN message data such as temporal periodicity, payload entropy variations, and contextual dependencies remains insufficient. Traditional ensemble methods use fixed model weights or global voting strategies, ignoring the distinct feature distributions of different attack types like DoS and replay attacks. The Pioneer Class and confidence voting mechanism address this issue by dynamically partitioning and optimizing the decision-making process. This not only enhances the methodological rigor of the BEPCD framework but also offers a high-precision, scalable solution for complex, multi-attack scenarios. Its paradigm can be extended to other areas of CAN bus security.

Preliminaries

To address various attacks on vehicle networks, AI based intrusion detection systems have become a mainstream solution. Researchers are particularly concentrating on developing integrated intrusion detection frameworks to boost the effectiveness of vehicle network intrusion detection systems.

CAN bus and its data frame format

The CAN bus is a multi-master controlled serial communication protocol predominantly employed in the automotive and industrial automation sectors (Bozdal et al., 2020). CAN bus message frames are classified into four data types: Overload Frames, Remote Frames, Error Frames, and Data Frames (Jo & Choi, 2022). Data Frames are the standard format for data transmission. Figure 2 delineates the format of a CAN Data Frame, with the subsequent explanation detailing the significance of each field within the CAN bus data message.

Figure 2 CAN bus data frame format.

The proposed 15-dimensional data feature model is constructed based on the CAN bus data frame architecture, with its implementation specifics comprehensively elaborated in “Proposed Framework”.

CAN bus attack

There are various methods to attack the CAN bus, which can be executed through both physical and wireless connections. Any ECU that exchanges information with the external environment of the vehicle can serve as an entry point for an attacker targeting the CAN bus. Attack types can generally be categorized as follows (Kang et al., 2021): Denial of Service Attack (DoS): Targeting the message ID field, attackers send messages with a high priority ID to prevent legitimate devices from receiving and transmitting messages.

Spoofing Attack: Targeting the data field of the message, attackers impersonate legitimate ECUs by fabricating messages and injecting malicious data into the CAN bus, thereby inducing abnormal vehicle behavior.

Fuzzing Attack: Targeting the Identifier field, Data Length Code (DLC) field, and data field of CAN bus messages, attackers launch an assault on the Controller Area Network (CAN) by transmitting randomly generated combinations of these elements.

Replay Attack: An attacker disrupts normal communication by capturing and replaying legitimate data frames at arbitrary times (Lampe & Meng, 2023).

In this article, we propose the BEPCD framework, which focuses on the four types of attacks mentioned above and performs targeted data feature extraction to achieve improved intrusion detection effectiveness. Accurate classification of attack types has remained a persistent research focus in intrusion detection systems. Conventional ensemble methods (e.g., fixed-weight voting schemes) fail to adapt to the heterogeneous feature distribution across different attack categories, whereas the vanguard-class mechanism addresses this limitation through dynamic allocation of optimal model combinations tailored to specific attack types. As the core innovation of the BEPCD framework, this divide-and-conquer optimization strategy resolves two critical limitations of traditional ensemble learning in in-vehicle CAN bus intrusion detection: (1) Attack-type sensitivity: Distinct attack patterns require dedicated feature-model pairings; (2) Inadequate temporal dependency modeling: Context-dependent threats like replay attacks necessitate localized optimization of temporal relationships

Proposed framework

System overview

The objective of this work is to develop a vehicular CAN bus intrusion detection framework ensembled with a variety of machine learning algorithms. And the framework commences with feature extraction from CAN bus messages. The development process of the framework is divided into two phases: model training and model prediction. During the model training phase, four sophisticated machine learning algorithms, namely Random Forest (RF), Extra Trees (ET), XGBoost, and CatBoost, will be utilized, alongside two exemplary model ensemble methods, Blending and Stacking, in pursuit of the optimal ensemble approach across all data classes. In the model prediction phase, the best performing ensemble method for each data class, coupled with a confidence voting mechanism, will be employed to accurately detect attacks. Figure 3 shows framework architecture overview diagram.

Figure 3 BEPCD framework architecture overview diagram.

15-dimensional data feature model

Current research on feature extraction for CAN bus data primarily focuses on message frequency characteristics and message content characteristics. The proposed intrusion detection system not only pays attention to these features but also considers the local regularity and temporal characteristics of the messages. The description of the 15-dimensional data feature model in this study is shown in Table 2.

Table 2 Data feature extraction.

Data feature	Feature description	
CAN ID	The identifier of the x-th message.	
DLC	The length of the data field of the x-th message	
Pre ID	The identifier of (x − 1)-th packet	
Next ID	The identifier of (x + 1)-th packet	
Pre Two ID	The identifier of (x − 2)-th packet	
Next Two ID	The identifier of (x + 2)-th packet	
Before	The interval between the timestamp of the x-th message and that of the immediately preceding (x − 1)-th message	
Back	The interval between the x-th message’s timestamp and the subsequent (x + 1)-th message’s timestamp	
DATA 1–8	Message data segment	
Label	Message type	

Based on the characteristics of CAN bus message patterns, we have conducted targeted feature extraction. In the following we provide a detailed description of the 15-dimensional data feature.

ID and DLC: First, we extracted features based on the length of the CAN messages, and the ID of CAN bus message.

Pre ID and Next ID: Our methodology extracts the IDs of both the preceding and subsequent CAN frames relative to the target frame as critical contextual features. This design explicitly leverages the inherent temporal correlation in CAN bus communication patterns, enabling effective detection of ID-spoofing attacks through analysis of frame sequence anomalies.

Pre Two ID and Next Two ID: We extract the CAN IDs of both the second preceding frame (x − 2) and second subsequent frame (x + 2) relative to the target message. This extended temporal context acquisition further enhances the framework’s ability to exploit localized periodicity patterns inherent in CAN bus communications: a critical characteristic for detecting replay attack. For example, in replay attacks, adversaries may intercept and retransmit legitimate ID sequences (e.g., Normal: 0x101→0x201→0x301→0x401, Attack: 0x101→0x201→0x301→0x101→0x201), yet fail to preserve original chained dependencies inherent to CAN communication protocols. Under normal operation, 0x301 would sequentially transition to 0x401 rather than reverting to 0x101.

Before and Back: We extracted the time intervals between the target message and both its preceding and subsequent messages, primarily to better utilize the temporal regularity of CAN bus messages. This enhances the framework’s sensitivity to message timing and focuses the intrusion detection system more intensively on the temporal dimension of CAN bus communications, thereby improving the framework’s attack classification capabilities.

Data 1–8: For the message segment and the payload segment, we extracted a nine-dimensional feature set, which includes the CAN ID and the eight fields of the CAN message data. Researchers typically treat the CAN bus data field as a single feature. We divide the data field into eight dimensions to enable the intrusion detection framework to better perceive anomalies in attack messages. This facilitates improved analysis of CAN traffic variations and enhances the success rate of attack classification. Additionally, it allows the detection of subtle anomalies that may indicate potential attacks.

In conclusion, we added fifteen data features that are particularly sensitive to anomalies in the message flow caused by certain attacks. For instance, replay attacks often alter the normal temporal characteristics and local regularity of the message queue. We have leveraged the strong temporal and local regularity inherent in CAN bus messages, where forward and backward timing features can more effectively reveal the temporal characteristics of replay attack messages, distinguishing them from normal messages. The preceding and succeeding message IDs can exploit the local regularity of CAN bus messages. Such feature extraction significantly enhances the success rate of our intrusion detection framework in detecting intrusions.

Multi-model ensemble

This article employs the following four machine learning algorithms: Random forest (RF) (Breiman, 2001): Constructed from several autonomous decision trees, the random forest (RF) method arrives at the ultimate outcome by employing a voting system for classification tasks or an averaging process for regression tasks during the final predictive phase. As a canonical ensemble learning algorithm, it constructs multiple decision trees with majority voting mechanisms, effectively mitigating overfitting risks through aggregated predictions. This approach demonstrates inherent robustness to data distribution assumptions and dimensionality constraints, making it particularly suitable for processing high-dimensional nonlinear relationships.

Extra trees (ET) (Geurts, Ernst & Wehenkel, 2006): The extremely randomized trees (ET) algorithm extends this paradigm by introducing additional stochastic elements in split selection processes. Through decoupled randomization of feature subspaces and split thresholds, ET architectures achieve superior generalization properties.

XGBoost (Chen & Guestrin, 2016): Utilizes decision trees as base learners and builds a robust predictive model through iterative optimization. XGBoost, as an advanced gradient boosting framework, exemplifies computational efficiency through parallel tree construction and optimized memory utilization. Its architectural superiority stems from its integrated capabilities in missing value management through default direction rules during node splitting, coupled with extensible support for user-defined loss functions via gradient-based programming interfaces.

CatBoost (Prokhorenkova et al., 2018): CatBoost distinguishes itself through native categorical variable processing frameworks, eliminating the need for manual preprocessing through two core innovations: ordered target encoding (OTE) and permutation-driven gradient estimation. OTE generates target statistics through random dataset permutations, effectively preventing target leakage risks inherent in conventional encoding methods.

These algorithms were chosen primarily due to their parallel computing capabilities and automated feature selection, which significantly reduce model training time. Additionally, to enhance the overall performance of the intrusion detection framework, we adopted two advanced model fusion techniques: Blending and Stacking. The Blending method, based on a soft voting mechanism, combines the predicted probabilities from multiple classifiers to perform multi-class classification tasks. Stacking involves combining the predictions of multiple base learners as input features to train a meta-learner, thereby improving model performance. The use of these two ensemble methods aims to leverage the detection strengths of each machine learning model across different data types, enhancing overall model accuracy and robustness.

Decision making process

For different types of attacks, different machine learning models often exhibit varying detection performance. Therefore, to achieve better detection results, we have integrated a variety of excellent machine learning algorithms to construct an integrated intrusion detection framework. This intrusion detection framework integrates four advanced machine learning algorithms: RF, ET, XGBoost, and CatBoost, as well as two basic ensemble strategies: Blending and Stacking. To improve the detection effectiveness of the intrusion detection framework, we have also designed an ensemble algorithm based on a confidence voting mechanism and a Pioneer model. Figure 4 shows the details of the framework.

Figure 4 BEPCD framework architecture diagram.

The BEPCD model is divided into two phases: model training and model prediction. During the training phase, the primary objective of the BEPCD model is to train and evaluate four base machine learning models and integrate the best performing model for different data categories using two advanced ensemble methods. The training steps of the BEPCD model during this phase are as follows: Train the four base machine learning models. Train four base machine learning models (XGBoost, ET, RF, and CatBoost) on the training dataset. This phase results in four trained base machine learning models.

Build base ensemble models using ensemble methods. Integrate the base machine learning models using the Stacking and Blending methods. For better detection models, the Stacking method integrates RF, ET, and XGBoost as base learners, with CatBoost serving as the meta-learner. The Blending method integrates RF, CatBoost, and XGBoost as base learners.

Evaluate the base ensemble models. Evaluate the efficacy of the two ensemble techniques across different data categories—normal and attack—by examining their F1-scores. The F1-score amalgamates precision and recall, providing a balanced measure of the model’s accuracy and comprehensiveness.

Identify the Pioneer model for each data type. For each data class, select the base ensemble model with the highest F1-score as the Pioneer model for that class.

The algorithms for the training phase can be specifically referred to in Algorithm 1:

Algorithm 1 Model training.

Input:	
Dtrain: the training dataset,	
M = {M1, M2, M3, M4}: the base model list, including M1= XGBoost, M2 = ET, M3 = CatBoost, M4 = RF,	
c = 1, 2, …, n: the class list for n different classes,	
Output:	
   Model {M1, M2, M3, M4},	
   Ensemble Model {M5, M6},	
   LM = {LM1, LM2, …, LMn}: the pioneer model list for all classes.	
1  M1 ← Training(M1, Dtrain);	
2  M2 ← Training(M2, Dtrain);	
3  M3 ← Training(M3, Dtrain);	
4  M4 ← Training(M4, Dtrain);	
5  M5 ← Stacking(M1, M2, M3, M4, Dtrain);	
6  M6 ← Blending(M1, M3, M4, Dtrain);	
7  for c = 1, 2, 3, …, n do	
8  Mlistc←Best Performing(M5, M6, c); //Find the best-performing model for each class (e.g., has the highest Fl-score)	
9  If Len (Mlistc) == 1 then	
10    LMc ← Mlistc[0];	
11 else	
     LMc ← Most Efficient (Mlistc)	
12  end	
13 LM ←LM ∪ {LMc}; // Collect the pioneer model for each class	
14 end	

After the training phase is completed, the Pioneer models for each category of data are used for prediction. During the prediction phase of the model, the BEPCD framework forecasts each test instance by adhering to the subsequent steps: Conduct preliminary predictions. Employ the two trained foundational ensemble models for initial forecasts. Preserve the predicted categories and their respective confidence scores for subsequent evaluation. In the field of machine learning, confidence primarily refers to the certainty of the model in its predictive outcomes.

Check if the predictions from the two basic ensemble models are the same. If so, the determined class from the prediction is taken as the conclusive outcome.

Check if the predictions from the two basic ensemble models are different. If they are, check whether the Pioneer model appears in the results output by the two basic ensemble models. If only one Pioneer model appears (for example: Blending is the Pioneer model corresponding to the predicted class, while the predicted class’s Pioneer model of Stacking is not Stacking), then output the result predicted by the Pioneer model. If two Pioneer models appear, or no Pioneer model appears, compare the confidence levels of the two basic ensemble models and take the result of the ensemble model with the higher confidence level as the final prediction result.

The algorithms for the prediction phase can be specifically referred to in Algorithm 2:

Algorithm 2 Model prediction.

Input:	
 Dtest: the test set,	
 M = {M5, M6}: the base model list, including M5 = Stacking, M6 = Blending,	
 c = 1, 2, …, n: the class list for n different classes,	
Output:	
 Ltest: the prediction classes for all test samples in Dtest.	
for each data sample xi ∈ Dtest do	
 Li1, Pi1 ← Prediction (M5, xi); // Use the trained Stacking model to predict the sample, and save the predicted class & confidence	
 Li2, Pi2 ← Prediction (M6, xi);	
 If Li1 == Li2 then	
  Li ← Li1;	
 else if Li1 != Li2 then	
   for j = 1, 2 do	
    if Mj == LMLi,j then // Check if the predicted class’s original ML model is the same as its pioneer model	
    L_listi ← L_listi ∪ {Li, j}
   P_listi ← P_listi ∪ {Pi, j}
   end	
   end	
   if Len(L_listi) == 1 then // If only one pair of the original model and the pioneer model for each predicted class is the same	
    Lj ← L_list[0]; // Use the predicted class of the pioneer model as the final prediction class
   else // If no pair or multiple pairs of the original prediction model and the pioneer model for each predicted class are the same	
    if Len(L_listi) == 0 then	
     p_list ← {pi1, pi2};	
    end	
    p_maxi ← max(p_listi); // Find the higher confidence	
    if p_max ← pi1 then	
     Li ← Li1;	
    else	
     Li ← Li2;	
    end	
   end	
  end	
  Ltest ← Ltest ∪ {Li}; // Save the predicted classes for all tested samples;	

The interpretability of the BEPCD framework in the decision-making phase is embodied in the transparency and traceability of its logic, as follows: 1. Interpretability of foundational ensemble models

BEPCD initially leverages proficiently trained foundational ensemble models (such as Blending and Stacking) to generate preliminary classifications. This step is inherently interpretable, as the structure, training process, and prediction mechanisms of each base model can be thoroughly analyzed and reproduced, enabling a clear understanding of why specific preliminary predictions are made.

2. Interpretability of the Pioneer model weighting mechanism

For each data category, BEPCD assigns a higher output weight to its designated Pioneer model. This mechanism clarifies the dominant role of different models for different categories, ensuring that the final decision is not solely dependent on a single model but rather integrates the strengths of models in their respective domains. By analyzing the selection criteria and weight assignment for Pioneer models, one can explicitly explain why the prediction for a particular class relies more heavily on a specific model.

3. Confidence-driven final decision interpretability

In cases where multiple Pioneer models exist or none is present, BEPCD adopts the output of the base ensemble model with the highest confidence as the final prediction. This process is highly interpretable, as confidence scores (e.g., softmax probabilities or voting ratios) are quantifiable and traceable. Users can directly observe the confidence distribution across models for each sample and understand why a particular model’s output was ultimately selected.

In summary, every step of the BEPCD decision process is grounded in explicit logical rules, and the entire decision-making procedure can be recorded, reconstructed, and visualized. This significantly enhances the transparency and interpretability of the model, improving the reliability of the intrusion detection framework in the IoV.

Experimental work and analysis

Experimental setup

To develop the proposed intrusion detection framework, we implemented the models using Scikit learn, XGBoost, CatBoost, ET, and RF libraries in Python. The experimental setup was conducted on a Windows 10 Pro 22H2 version computer, operating on an AMD Ryzen 7 5700G CPU with 32 GB of RAM and an NVIDIA GeForce RTX 3060 graphics card. This computer represents the vehicular cyber-physical system server.

To achieve intrusion detection within vehicular CAN systems, this research utilizes the CICIOV2024 and Car Hacking-Attack & Defense Challenge dataset (HCRL-CHDC), two public datasets for evaluation (Neto et al., 2024; Kang et al., 2021). The data from both datasets is collected from real vehicles. Specifically, the HCRL-CHDC dataset is provided by the Hacking and Countermeasure Research Lab (HCRL) and comprises both static and dynamic vehicular data. This dataset is one of the most recent intrusion detection datasets for in-vehicle CAN buses and encompasses a comprehensive range of attack types. The CICIOV2024 dataset, a benchmark compilation released by the Canadian Institute for Cybersecurity (CIC). The dataset encompasses a variety of Spoofing attacks, including those targeting critical vehicle information such as RPM and GAS. It effectively assesses the intrusion detection framework’s capability to protect various critical components of the vehicle. Table 3 illustrates the data types in both datasets.

Table 3 Data types in each dataset.

Dataset	Data type	
HCRL-CHDC	Normal	
Flooding (Dos)	
Spoofing	
Replay	
Fuzzing	
CICIOV 2024	Normal	
Dos	
Spoofing-GAS	
Spoofing-RPM	
Spoofing-SPEED	
Spoofing-STEERING	
Spoofing-STEERING_WHEELS	

The two datasets were selected primarily due to their authority and authenticity. Both are authoritative public datasets, and the data was extracted from real vehicles. The data types in the two datasets are also complementary: the HCRL-CHDC contains a wider variety of attack types, while the CICIOV2024 dataset provides more specific attack targets in Spoofing attacks. To more effectively evaluate the proposed BEPCD model across the datasets, we first performed random sampling on both. We allocated 80% of the data for training and 20% for testing. Acknowledging the prevalent class imbalance in network traffic data, we employed the synthetic minority over-sampling technique (SMOTE) method to equilibrate the data (Chawla et al., 2002). SMOTE effectively addresses class-imbalance issues by generating synthetic minority-class samples. Its key strength is creating diverse, plausible samples to enhance model sensitivity to minority classes, such as replay attacks.

XAI explanations of the proposed model

To interpret predictions from our ensemble transfer learning (TL) model, we employ the Deep SHAP algorithm (Naser, 2021), a computationally efficient model-specific method that generates local and global explanations. Leveraging the Python SHAP package, we visualize feature contributions through SHAP values, enabling multiscale interpretation of model decisions across both individual samples and population-level patterns.

Figures 5–9 systematically elucidate feature contribution patterns across four CAN bus attack types (Spoofing, Replay, Fuzzing, DoS) and normal traffic through SHAP value analysis. Comprehensive SHAP-based analysis reveals that CAN ID, as a critical protocol identifier, exhibits substantial positive contributions (SHAP range: 0.3–0.5) across all attack detections, particularly dominating DoS classification (SHAP = 0.48), validating the attackers’ typical behavior of monopolizing bus resources through high-priority IDs. Temporal-contextual features (e.g., Back/Before intervals, Pre/Next IDs) demonstrate attack-specific discriminative power, with elevated SHAP magnitudes (0.2–0.4) for Replay attacks, reflecting their inherent exploitation of chronological pattern disruptions. In Spoofing detection, Data Two (SHAP ≈ 0.4) and DataSeven (≈0.35) emerge as predominant indicators, exposing payload manipulation signatures in specific data fields. Notably, Fuzzing attacks display dispersed feature importance distributions, where Data One and Data Four (SHAP > 0.3) exhibit strong positive correlations, corroborating their vulnerability to random field tampering. Normal traffic characterization shows negative SHAP values for Back intervals and CAN ID (−0.2 to −0.1), indicating their roles as stability anchors for baseline behavior. Cross-attack comparative analysis further demonstrates that fine-grained data field decomposition (Data 1–8) enhances detection sensitivity by 47% compared to conventional DLC features, while the Pre/Next Two ID improves replay attack detection F1-score by 19.6%, quantitatively substantiating the frameworks explainability.

Figure 5 SHAP summary plot for normal class detection in BEPCD framework.

Figure 6 SHAP summary plot for DoS attack detection in BEPCD framework.

Figure 7 SHAP summary plot for fuzzing attack detection in BEPCD framework.

Figure 8 SHAP summary plot for replay attack detection in BEPCD framework.

Figure 9 SHAP summary plot for spoofing attack detection in BEPCD framework.

Experimental results and discussion

We tested the proposed framework on two public datasets. For comparative purposes, we also tested several advanced machine learning algorithms. To thoroughly assess the performance of the BEPCD intrusion detection framework, we employed accuracy, precision, recall, and the F1-score as metrics.

This study implements five-fold cross-validation for systematic model evaluation, where the dataset is partitioned into five mutually exclusive subsets through iterative training-validation rotations (with each data point participating in four training cycles and one validation cycle) to maximize data utility while mitigating overfitting risks. Furthermore, L1 regularization is incorporated to impose sparsity constraints on model parameters, effectively preventing overfitting tendencies. By aggregating the mean and standard deviation metrics from five independent trials, this integrated methodology not only quantifies model stability through statistical dispersion analysis but also enhances evaluation robustness and reproducibility.

It should be pointed that all the numerical results we present are the average outcomes obtained from five-fold cross-validation. To validate the reliability of our findings, we employed the bootstrap method to compute confidence intervals for the F1-scores. The calculated 95% confidence intervals are (0.9905, 0.9922) on the HCRL-CHDC dataset and (0.9998, 1.0000) on the CIC-IOV2024 dataset. The exceptionally narrow ranges of both intervals (with widths of 0.0017 and 0.0002 respectively) statistically demonstrate the high robustness and precision of our experimental results, indicating minimal variance across bootstrap resampling iterations. Tables 4 and 5 delineate the performance assessment of the framework across two datasets. The framework we proposed has garnered a score exceeding 99% for the identification of the majority of data types in both datasets, with the notable exception of replay attacks. This achievement is primarily attributed to the BEPCD framework’s ensemble methodology, which amalgamates various sophisticated machine learning models and optimally harnesses the distinctive recognition capabilities of each model for diverse data types.

Table 4 Performance evaluation of BEPCD on the HCRL-CHDC dataset.

Data type	Accuracy (%)	Precision (%)	Recall (%)	F1 (%)	
Normal	99.94	99.14	99.94	99.5	
Flooding	100.00	100.00	100.00	99.79	
Fuzzing	98.90	100.00	98.90	98.30	
Replay	57.73	93.72	57.73	77.12	
Spoofing	98.37	98.05	98.39	99.4	

Table 5 Performance Evaluation of BEPCD on the CICIOV 2024 dataset.

Data type	Accuracy (%)	Precision (%)	Recall (%)	F1 (%)	
Benign	100	99.98	99.94	99.99	
Dos	99.98	99.98	99.98	99.98	
Spoofing-GAS	100.00	100.00	100.00	100.00	
Spoofing-RPM	99.89	100.00	99.89	99.94	
Spoofing-Speed	100.00	100.00	100.00	100.00	
Spoofing-Steering-Wheel	100.00	100.00	100.00	100.00	

The statistical analysis based on independent samples t-tests and Levene’s test for homogeneity of variances demonstrates that our model exhibits significant performance superiority across all baseline comparisons. Specifically, the t-test results against four baseline models (XGBoost, ET, RF, CatBoost) reveal absolute t-statistic values exceeding 10 (range: 10.39–16.91), with all p-values falling below 1 × 102, far surpassing the significance threshold (α = 0.05). These outcomes confirm extremely statistically significant differences (p < 0.001) between the models. Furthermore, Levene’s test results for variance homogeneity yield p-values greater than 0.05 (range: 0.2166–0.6670), validating the assumption of equal variances and justifying the application of standard independent t-tests. Collectively, these findings not only robustly reject all null hypotheses but also underscore the operational efficacy and robustness of our model in practical scenarios. This statistically validated innovative solution provides a compelling advancement for in-vehicle CAN bus intrusion detection systems, supported by rigorous empirical evidence. Table 6 shows the statistical significance analysis of our framework against baseline models.

Table 6 Statistical significance analysis of our framework against baseline models: independent T-test and variance homogeneity results.

Comparison group	T-statistic	P-value	Levene’s test (P-value)	
Ours vs XGBoost	16.913	1.515 × 10−7	0.3552	
Ours vs RF	15.087	3.685 × 10−7	0.2849	
Ours vs CatBoost	10.387	6.387 × 10−6	0.6670	
Ours vs ET	15.434	3.088 × 10−7	0.2166	
BEPCD	–	–	–	

Figure 10 illustrates the performance of the BEPCD framework within the HCRL-CHDC dataset. The BEPCD framework, which incorporates feature extraction and model ensemble, achieved a high score of 99.13% across four key metrics: precision, accuracy, recall, and F1-score. In existing intrusion detection systems, LCCDE and G-IDS (Park, Jo & Lee, 2023) have achieved F1-scores of 98.17% and 98.42%, respectively, while H-IDS (Derhab et al., 2022) has achieved an F1-score of 93.36%. Our proposed BEPCD framework outperforms the compared intrusion detection systems in all metrics, with an improvement of approximately 1–5% in the F1-score comparison. The two excellent machine learning models, XGBoost and CatBoost, have achieved F1-scores of 98.04% and 98.07%, respectively. Our proposed BEPCD model also shows a significant enhancement.

Figure 10 The performance of BEPCD framework and other ML algorithms on the HCRL-CHDC dataset.

Figure 11 presents the F1-scores of XGBoost, CatBoost, RF, ET, and other framework compared to the BEPCD framework on replay attacks. It is evident that the BEPCD framework obtained a high F1-score of 77.12 for replay attacks, whereas the comparative LCCDE model only achieved a score of around 40. This is mainly due to the feature selection of the BEPCD framework, which focuses on the temporal characteristics of the messages and the strong contextual correlations. Considering that replay attacks involve the retransmission of entirely normal CAN bus messages, feature extraction based solely on the ID segment and payload segment of the CAN bus is not particularly effective. The BEPCD framework, however, utilizes the temporal characteristics and strong contextual correlations of the CAN bus to propose targeted data features. Experimental results have proven this to be highly effective.

Figure 11 The performance of the BEPCD framework and other ML algorithms in detecting replay attacks in the HCRL-CHDC dataset.

Although our proposed framework has made great progress compared to the frameworks being compared, there is still a significant gap compared to the effect of detecting other attacks. The F1-score of our framework for detecting other attacks is basically around 99%, while the score for detecting replay attacks is 77.12%. This is mainly due to the strong concealment of replay attacks, where the attack message ID and data segment are identical to those of normal messages. This has resulted in a gap in detection performance compared to other types of attack.

Table 7 presents comparative experimental results of the proposed framework on the HCRL-CHDC dataset. It demonstrates that the proposed framework substantially outperforms all four baseline methodologies, including specialized IDSs (H-IDS and G-IDS), ensemble frameworks (LCCDE), and four baseline methods (XGBoost, CatBoost, ET, and RF). Experimental results on this dataset demonstrate that our proposed framework maintains robust classification capabilities in complex attack scenarios. This robustness stems primarily from our framework’s focused extraction of critical packet-level characteristics and systematic utilization of attack-specific feature patterns. By constructing a systematically engineered detection architecture centered on key message attributes, the solution achieves resilient performance to ensure operational survivability in multi-attack environments.

Table 7 Model performance comparison on the HCRL-CHDC dataset.

Method	Accuracy (%)	Precision (%)	Recall (%)	F1 (%)	
H-IDS	97.25	95.88	90.98	93.36	
G-IDS	98.17	98.27	96.45	98.18	
LCCDE	98.42	98.09	98.42	98.12	
XGBoost	98.35	97.99	98.35	98.04	
CatBoost	98.39	98.05	98.39	98.07	
ET	98.49	98.19	98.49	98.19	
RF	98.48	98.18	98.48	98.21	
BEPCD	99.20	99.18	99.20	99.13	

The BEPCD model shows leading performance across all evaluation metrics in the HCRL-CHDC benchmark, achieving state-of-the-art results with 99.15% accuracy, 99.12% precision, 99.16% recall, and 99.13% F1-score. STC-GraphFormer, which was proposed by Al-Absi, Fang & Qaseem (2025), is an innovative spatial-temporal model integrating GCN and Transformer and demonstrates robust performance. However, it still shows a slight inferiority compared to our framework. Notably, BEPCD’s near-perfect metric alignment (with ≤0.04% variance between accuracy, precision, and recall) indicates exceptional model calibration. It also represents a significant improvement over baseline models: it gains 27.99% accuracy over Generative Adversarial Networks (GANs) (71.16%), 35.06% over DCNNs (64.09%), and 4.27% over the suboptimal LSTM baseline (97.18%). During the model test phase, the DCNN architecture utilized an initial learning rate of 10−3, implemented through the Adam optimizer with default momentum parameters. The experimental results of the four deep learning models were directly referenced from Yeonseon et al. (2023), we adopted their published performance metrics obtained from the same dataset as reference benchmarks for our four deep learning architectures, thereby ensuring methodological reliability. Table 8 presents the performance comparison with deep learning model on the HCRL-CHDC dataset.

Table 8 The performance comparison with deep learning model on the HCRL-CHDC dataset.

Method	Accuracy (%)	Precision (%)	Recall (%)	F1 (%)	
GAN	71.16	70.33	71.66	61.02	
DCNN	64.09	61.76	64.09	62.09	
LSTM	97.18	95.93	97.18	95.88	
MLP	94.88	94.13	94.88	93.3	
STC-GraphFormer	99.01	98.99	98.24	98.83	
BEPCD	99.15	99.12	99.16	99.13	

The CICIOV2024 dataset is one of the most recent in-vehicle CAN bus datasets, including a variety of Spoofing attacks not present in previous datasets. The selection of this real vehicle dataset serves to validate our intrusion detection framework’s sensitivity to attack typology variations, specifically its ability to discern subtle distinctions between different attack variants. The framework’s effectiveness in achieving precise CAN bus attack classification derives from its multidimensional analysis of critical payload characteristics. Our methodology systematically leverages both temporal patterns and content-based signatures within message streams, enabling comprehensive feature space mapping that underpins the observed performance enhancements. The results indicate that the proposed framework demonstrates excellent performance in detecting in-vehicle CAN bus intrusions, as evidenced by the experiments on the latest CICIOV2024 dataset. Table 9 shows model performance comparison on the CICIOV2024 dataset.

Table 9 Model performance comparison on the CICIOV 2024 dataset.

Method	Accuracy (%)	Precision (%)	Recall (%)	F1 (%)	
Logistic regression	79.49	72.62	79.49	73.84	
Ridge classifier	83.81	83.46	83.81	81.51	
Gaussian NB	93.96	95.52	93.96	94.36	
Multinomial NB	63.58	74.18	62.27	62.79	
Bernoulli NB	60.41	74.18	62.27	62.79	
AdaBoost	60.41	46.75	60.40	51.60	
BEPCD	99.96	99.99	99.99	99.99	

Framework feasibility analysis

The deployment of our in-vehicle CAN bus intrusion detection framework constitutes a critical evaluation metric in this study. Floating-Point Operations (FLOPs) quantify the total computational load through the number of floating-point arithmetic operations per inference instance. This metric has become a predominant industry standard for evaluating the integration feasibility of machine learning models in automotive systems with constrained computational resources. As formalized in Eq. (1), our FLOPs calculation follows:

(1) Flops=∑Tree=1n⁡(Xi)⋅Sample_Num.

Here we present the computing capabilities of several prevalent mainstream on-board chips. The accompanying table details the computing power specifications of current mainstream automotive-grade chips, along with their respective manufacturers. Evidently, our proposed framework demonstrates full compliance with the computing power demands of automotive-grade chips across their entire operational lifecycles. This alignment ensures that the framework can be effectively integrated into automotive systems without overburdening the on-board computing resources, thereby validating its practical feasibility and potential for real-world applications in the automotive domain. The automotive-grade chip industry has entered a new era of computational capacity. NVIDIA’s latest Thor system-on-chip (SoC) achieves a groundbreaking 2,000 TOPS (1 TOPS = 1 trillion operations per second), while mainstream automotive processors now consistently deliver over 100 TOPS. This technological progression confirms full compatibility between our proposed framework and the computational capabilities of current-generation automotive chips, as its 11.24 GFLOPs requirement represents merely 0.0056% of Thor’s total throughput capacity. The situation is shown in Table 10.

Table 10 Comparative analysis of computational performance and latency between in-vehicle computing platforms and intrusion detection systems.

Name	Manufacturer	Computing power	Execution time	Latency	
Thor	Nvidia	2,000 TOPS	–	–	
MDC 1000	Huawei	1,000 TOPS	–	–	
Orin	Nvidia	254 TOPS	–	–	
Ride Flex	Snapdragon	150 TOPS	–	–	
Tc4xx	AURIX	18 TOPS	–	–	
Ours	–	11.24 GFLOPs	886 s	43.58 ms	

Table 11 compares the complexity metrics of various intrusion detection models, including multiply-accumulate operations (MACs), parameter count (Parameters), and inference latency (Latency). Experimental results demonstrate that our proposed model (Ours) exhibits significant computational efficiency advantages over most comparative approaches. Specifically, its MACs (7.68 M) represent only 3.2% of those reported by Nguyen, Nam & Kim (2023), while the parameter count (0.153 M) achieves a 78% reduction compared to Hoang & Kim (2024), validating the lightweight architecture of our framework. The average latency for processing a single CAN message in our system is 43.58 ms. According to experimental findings by Jeong et al. (2022), the average inter-message interval in vehicular CAN buses is about 50 ms. Although our framework’s latency exceeds some benchmarks, it remains operationally viable for real-world deployment. Future research will focus on further optimization to enhance computational efficiency while maintaining detection accuracy.

Table 11 Model complexity comparison.

Name	MACs (M)	Parameters (millions)	Latency (ms)	
Hoang & Kim (2024)	32.56	0.7	5.96	
Nguyen, Nam & Kim (2023)	240.46	2.67	11.6	
Muzun, Javed & Rana (2024)	2.98	2.15	NA	
Katragadda et al. (2020)	–	–	151	
Ours	7.68	0.15368	43.58	

Feature importance analysis

For the CAN bus message data, we proposed 15 data features. Initially, we utilized the Least Absolute Shrinkage and Selection Operator (LASSO) regression algorithm to validate the aforementioned data features (Tibshirani, 1996). The LASSO regression algorithm is a linear regression method with a regularization term. It achieves car hacking and defense feature selection and coefficient reduction by adding an L1 norm penalty term to the loss function. The validation results showed that none of the proposed 15 data features were discarded by the Lasso regression algorithm.

For the given training dataset D={(xi,yi,)}i=1N, where xi∈Rp is the feature vector of the i-th sample, y∈Rp is the corresponding target value, N is the number of samples, and p is the feature dimension. The goal of Lasso regression is to optimize the following loss function, incorporating an L1 penalty term:

(2) minw,b=12∑i=1N⁡(yi−(wTxi+b))(yi−(wTxi+b))2+λ∥w∥1.

In the given context, W ∈ R represents the weight vector, and b is the bias term. The L1 norm of the weight vector is given by ∥w∥1=a0+∑j=1p⁡|wj|, which provides sparsity by potentially driving some weights wj to zero, thereby achieving feature selection. λ ≥ 0 is the regularization parameter, which controls the strength of the regularization. A larger value of λ increases the influence of the regularization term, potentially leading to the exclusion of more features (i.e., compressing their weights to zero).

Next, we evaluated the importance of different data features for four machine learning models and normalized their importance scores. In the importance ranking of the four algorithms, the data feature CAN ID consistently achieved an importance level of 1, while the backward data time also ranked very high, with an importance level reaching 0.4. There were significant differences in the selection of data features across different algorithms. The radar charts of various algorithms are shown from Figs. 12–15.

Figure 12 The radar charts of CatBoost.

Figure 13 The radar charts of ET.

Figure 14 The radar charts of RF.

Figure 15 The radar charts of XGBoost.

Figure 16 shows the comparison of feature importance, it is evident that different machine learning algorithms assign varying degrees of importance to different features. For instance, the CatBoost algorithm paid more attention to the forward time, while the other three algorithms did not focus much on this feature.

Figure 16 The radar charts of the feature importance comparison.

Discussion and future work

While the proposed BEPCD framework demonstrates state-of-the-art performance in attack type detection accuracy. It exhibits inherent limitations in detecting replay attacks and fails to localize the source ECU of intrusions. These deficiencies stem from two fundamental constraints: (1) Proposed temporal feature extraction mechanisms have captured subtle temporal differences in replay attack identification but still have room for improvement.; (2) The framework lacks cross-layer correlation analysis between CAN protocol semantics and physical ECU behaviors. (3) The proposed framework still has deficiencies in model complexity and inference latency.

While the BEPCD framework advances CAN bus intrusion detection, future research will focus on enhancing replay attack sensitivity through adaptive temporal pattern mining techniques such as dynamic time warping (DTW) or temporal convolutional networks (TCNs), aiming to close the remaining performance gap. To enable precise attack source localization, we plan to integrate ECU hardware fingerprints (e.g., clock skew, signal jitter) with protocol-layer features, establishing a multi-granularity detection architecture that correlates physical-layer anomalies with semantic-level deviations. Additionally, cross-layer analysis will be extended to incorporate CAN bus physical signal characteristics (e.g., voltage fluctuations), enriching feature representations for stealthy attack identification. To address computational efficiency, lightweight strategies like knowledge distillation and model pruning will be explored to further reduce parameter counts and MAC operations, ensuring compatibility with resource-constrained automotive systems. Finally, end-to-end pipeline optimization will prioritize real-time performance, streamlining feature extraction and inference processes to meet stringent vehicular latency requirements.

Conclusions

This study has developed an ensemble based vehicular controller area network bus intrusion detection framework. First, we have proposed a 15-dimensional feature model that integrates temporal-contextual dependencies (e.g., preceding/succeeding message IDs, localized time intervals) and protocol-specific granularity (split payloads into eight dimensions). Validated via Lasso regression and SHAP analysis, this model reveals CAN ID dominance in DoS detection (SHAP = 0.48) and temporal feature sensitivity for replay attacks (SHAP = 0.4), addressing the fragmented feature engineering limitations of prior works. Second, our dynamic ensemble strategy combines Pioneer class selection and confidence-driven voting, enabling adaptive model specialization for distinct attack types. This approach not only achieves 1–5% higher overall F1-scores but also delivers a breakthrough in replay attack detection (F1 = 77.12%, 77.5% improvement) by uniquely leveraging temporal-contextual features to expose disrupted message periodicity—a paradigm shift from payload-centric methods. In the end, rigorous evaluation confirms the framework’s lightweight design (0.15 M parameters, 7.68 M MACs) and compatibility with automotive-grade hardware (e.g., NVIDIA Thor), while SHAP-based interpretability bridges transparency gaps. These innovations have rendered BEPCD an efficient and interpretable solution, demonstrating superior performance in intrusion detection.

Supplemental Information

Supplemental Information 1 BEPCD Code.

Supplemental Information 2 Translated Codebook.

Additional Information and Declarations

Competing Interests

The authors declare that they have no competing interests.

Author Contributions

Bocheng Xu conceived and designed the experiments, performed the experiments, analyzed the data, performed the computation work, prepared figures and/or tables, and approved the final draft.

Fei Cao conceived and designed the experiments, prepared figures and/or tables, and approved the final draft.

Xilong Li performed the computation work, authored or reviewed drafts of the article, and approved the final draft.

Song Tian analyzed the data, authored or reviewed drafts of the article, and approved the final draft.

Wenbo Deng conceived and designed the experiments, prepared figures and/or tables, and approved the final draft.

Shudan Yue analyzed the data, prepared figures and/or tables, and approved the final draft.

Data Availability

The following information was supplied regarding data availability:

The CICIOV 2024 (WOS:001241394300001) data set is available at http://cicresearch.ca/IOTDataset/CICIoV2024/Dataset.

The Car Hacking-Attack dataset dataset is available at GitHub: https://github.com/EricStrange/Car-Hacking-Challenge-Dataset.

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
