# Peer review of "BEPCD: an ensemble learning-based intrusion detection framework for in-vehicle CAN bus"

_PeerJ Computer Science, doi:10.7717/peerj-cs.3108_

## Round 0.1 · original submission · Major Revisions

Please respond to the comments from all 3 reviewers

Reviewer 1 ·

Basic reporting

Summary:
This paper proposed a basic ensemble and Pioneer Class Decision-based IDS for IoV. However, the paper lacks substantial contributions and presents poor writing and organization. I regret to reject this paper without further consideration.

Strengths:
1. The proposed topic is critical for improving the safety of in-vehicle CAN Bus systems.
2. The evaluation validates the effectiveness of the proposed method.

Weaknesses:

1. Lack of novelty: IDSs for the IoV have already addressed the challenges of feature extraction and complex attack classification. The proposed BEPCD, however, seems to be presented without a thorough review of existing works, resulting in a minimal contribution to the field.

2. Poor writing and readability: The manuscript quality is poor, with significant issues in the structure, presentation of formulas, and figures, making it difficult to read and comprehend.

Suggestions:
1. The introduction should avoid excessive background information. It is essential to highlight the paper's motivation and articulate its contributions concisely.

2. The authors should base their work on a more thorough review of related research. The proposed method currently does not present any notable advancement in the field. A clear and comparable competency matrix will highlight the value of the work.

3. Section 3 lacks details relevant to the paper’s contributions, such as the design rationale behind the method, an explanation of ensemble learning, and a clear description of Pioneer Class Decision.

4. The methods section is underdeveloped and lacks academic rigor. It is poorly structured and should be reorganized to meet academic writing standards.

5. The presentation of the evaluation results needs significant improvement. The results section should clearly outline the experimental setup, primary findings, ablation studies, and efficiency analysis.

Experimental design

See above

Validity of the findings

See above

Additional comments

N/A

Reviewer 2 ·

Basic reporting

The manuscript is well-written but contains some complex sentences that could be simplified for better readability. Proofreading for grammar and sentence structure is recommended.

The introduction provides sufficient background, but it should include more recent references on ensemble learning methods in cybersecurity.

Some figures lack clear descriptions. Ensure all figures are properly labeled and well explained in the text.

The dataset usage is well described, but it should be clarified if the raw data and code will be made publicly available for reproducibility.

Experimental design

The description of the BEPCD framework is detailed, but the rationale for selecting specific machine learning models (RF, XGBoost, ET, CatBoost) needs better justification.

The study uses two datasets, but more information on data preprocessing and handling of imbalanced data is needed.

The paper compares BEPCD with several models, but additional comparisons with deep learning-based approaches would strengthen the study.

The paper should provide more details on hyperparameter tuning and feature selection to enhance reproducibility.

Validity of the findings

The reported F1 scores are impressive, but statistical validation (e.g., confidence intervals) is missing.

There is no discussion on the robustness of BEPCD against adversarial attacks or real-world deployment scenarios.

The study does not provide insights into the computational cost of BEPCD compared to other methods.

Additional comments

The paper should discuss the scalability of BEPCD in large-scale CAN bus networks.

The conclusion should mention the study's limitations and propose directions for future research.

The novelty of BEPCD should be emphasized more clearly in contrast to existing ensemble learning techniques.

Reviewer 3 ·

Basic reporting

Strengths:
- The paper is written in professional and generally clear English.
- The introduction effectively contextualizes the importance of vehicle network security and CAN bus vulnerabilities.
- The literature review is comprehensive, covering various ensemble learning methods and their application to CAN bus intrusion detection.
- The structure aligns with standard research reporting norms.

Weaknesses and Suggested Improvements:
- Although the language is professional, there are occasional awkward phrases and grammatical issues. A thorough proofreading by a native English speaker is recommended. Examples:
Line 24: “With the rapid development and widespread adoption of Vehicles (IoV)”: Suggest: “With the rapid development and widespread adoption of Internet of Vehicles (IoV) technology.”
Line 33: “followed by the integration of multiple machine learning models to classify various data types using Pioneer class and confidence voting mechanisms”: Consider rephrasing for clarity.
- While the related works section is robust, a clearer differentiation between this work and past ensemble-based detection systems would strengthen the argument for novelty. And a brief explanation of why Pioneer class and confidence voting mechanisms were introduced would better set the stage for the methods section.
- Ensure that all claims are properly supported with citations. For instance, when stating that individual machine learning models struggle to detect diverse attack types (Lines 71-74), a citation to empirical evaluations supporting this claim would be beneficial.

These studies discuss the challenges individual models face in detecting complex CAN bus attacks and emphasize the role of hybrid and ensemble methods in improving detection performance.

Experimental design

Strengths:
- The research question is well-defined and addresses a critical gap in multi-attack intrusion detection for CAN bus systems.
- The use of ensemble learning, Pioneer class, and confidence voting mechanisms reflects methodological rigor.

Weaknesses and Suggested Improvements:
- Although Table 1 lists extracted features, there is a lack of detailed explanation regarding the rationale behind selecting specific features (e.g., why preceding and succeeding message IDs were included). Providing an example of how these features differentiate normal and malicious traffic would enhance clarity.
- Cross-validation or multiple train-test splits could improve the robustness of reported results, especially given the class imbalance issue. Consider reporting standard deviations along with mean performance metrics.
- While the detection accuracy is emphasized, computational efficiency (e.g., training time, prediction latency) is crucial for real-time vehicular applications. Including a performance comparison in terms of computation time would provide a more comprehensive evaluation. Emphasize computational efficiency through BCNN, which may offer a relevant benchmark here.
- The choice of XGBoost, CatBoost, Random Forest, and Extra Trees is reasonable but lacks justification based on prior evaluations in the automotive cybersecurity context. A brief rationale explaining their suitability would improve the methodology section. Recent ensemble-based and hybrid approaches, such as the study by [4] employing stacking-enriched learning with feature fusion, offer valuable comparative insights.

Validity of the findings

Strengths:
- The performance metrics (F1 score, precision, recall, and accuracy) are appropriate for the evaluation task.
- The results indicate that BEPCD outperforms existing frameworks on both datasets, particularly for replay attacks.

Weaknesses and Suggested Improvements:
- While the improvement from 40% to 71.45% in detecting replay attacks is notable, this is still relatively low compared to other attacks. Further exploration into why replay attacks remain challenging and how future iterations could address this would enhance the discussion section.
- The paper presents point estimates of evaluation metrics but does not assess statistical significance. Conducting statistical tests (e.g., paired t-tests) could strengthen the claim that BEPCD significantly outperforms baseline models.
- With an F1 score reaching 99.98% on the CICIOV2024 dataset, there is a potential risk of overfitting. Clarifying how overfitting was mitigated (e.g., through regularization, hyperparameter tuning) would bolster the credibility of the results. [1] highlights the importance of evaluating detection systems across real-world vehicle driving conditions, which could serve as a useful robustness benchmark.

Additional comments

More Suggested Enhancements:
- While the focus is on ensemble learning, deep learning-based intrusion detection systems are gaining traction in automotive cybersecurity. Discussing how BEPCD compares to recent deep learning approaches [1-5] would position the work within the broader landscape.
- Including a brief discussion on the feasibility of deploying BEPCD in real-world vehicular systems, considering hardware limitations and latency constraints, would improve the practical significance of the study.
- The paper mentions using public datasets, but providing a clear link to the specific versions of these datasets and the implementation code would enhance reproducibility.

---

## Round 0.2 · Major Revisions

Since one of the reviewers asked to reject the submission (without giving all the needed details), I had to ask for an additional review, which you can find below.

Please consider these comments when revising your submission.



1. Literature Review and Related Work

The current literature review does not sufficiently cover recent advancements in deep learning-based IDS for CAN bus security, particularly those demonstrating high accuracy and efficiency. Please consider incorporating and discussing the following (see the complete list below):

* Rai et al. (2025) evaluated deep learning methods, including LSTM, GRU, and VGG-16, achieving up to 99.89% accuracy in binary classification and 100% accuracy in multiclass classification on datasets like Car Hacking, Survival Analysis, and OTIDS.

* Agbo (2024) developed a hybrid IDS combining 1D-CNN with Bidirectional LSTM, achieving 99.64% accuracy on the ROAD dataset, highlighting the potential of hybrid deep learning models in capturing both spatial and temporal features of CAN traffic.

* Suwwan et al. (2021) implemented 1D CNN, LSTM, and GRU networks on a recent CAN attack dataset, with all models achieving an F1-score of near 1.0, showcasing the effectiveness of these architectures in intrusion detection.

* Devnath (2023) introduced GCNIDS, a Graph Convolutional Network-based IDS for the CAN bus, enhancing attack detection accuracy while minimizing the need for manual feature engineering.

* Althunayyan et al. (2024) proposed a robust multi-stage IDS using hierarchical federated learning, employing an artificial neural network and LSTM autoencoder to detect both known and novel attacks, achieving high F1-scores with low false alarm rates.

Please expand the related work section to include these studies, comparing their methodologies, datasets, and results with your proposed approach. This will contextualize your work within the current state-of-the-art and highlight its contributions.

2. Deep Learning Model Performance

The reported performance of DCNNs in your study is significantly lower than that documented in existing literature, raising concerns about the experimental setup and evaluation.

Please provide a detailed explanation for the underperformance of DCNNs in your experiments. Consider factors such as:

* Differences in dataset characteristics (e.g., size, class imbalance).

* Model architecture and hyperparameter settings.

* Training procedures and evaluation metrics.

* Compare your DCNN implementation with those in the referenced studies to identify potential discrepancies.

Please include an in-depth analysis of the DCNN performance in your results section, discussing possible reasons for the observed outcomes and how they align or differ from existing research.

Explainable Artificial Intelligence (XAI) Considerations

The manuscript lacks discussion on XAI approaches, which are increasingly important for understanding and trust in IDS, especially in safety-critical applications like automotive systems. Please consider integrating discussions on XAI methods relevant to IDS, such as:

Jeong et al. (2023) presented X-CANIDS, a signal-aware explainable IDS for CAN-based in-vehicle networks, improving intrusion detection performance and enabling understanding of which signal or ECU is under attack.

Saheed and Chukwuere (2024) proposed XAIEnsembleTL-IoV, an explainable AI ensemble transfer learning model for detecting zero-day attacks in the Internet of Vehicles.

Nwakanma et al. (2023) provided a review of XAI models used in intelligent connected vehicle IDSs, discussing their taxonomies and outstanding research problems.

Discuss how incorporating XAI techniques could enhance the interpretability and trustworthiness of your proposed framework.

Add a subsection on XAI in your discussion, outlining potential methods for integrating explainability into your IDS and the benefits thereof.

Lightweight and Real-Time IDS for Embedded Systems

The submitted study does not address the feasibility of deploying the proposed IDS in resource-constrained embedded automotive systems, where lightweight and real-time performance is crucial.

Evaluate the computational complexity and resource requirements of your framework. Compare with recent lightweight IDS approaches, such as:

* Althunayyan et al. (2024) introduced a lightweight, in-vehicle IDS leveraging deep learning algorithms, suitable for real-time detection systems.

* Altalbe (2023) proposed an accurate and low-complexity IDS for in-vehicle networks based on feature fusion and ensemble learning, enabling real-time attack detection with high accuracy.

Please include a performance evaluation section discussing the real-time applicability and efficiency of your IDS, considering the constraints of embedded automotive environments.

Clarification of Novelty and Contributions

While the BEPCD framework introduces novel components, the manuscript could better articulate its unique contributions in comparison to existing methods. Clearly delineate how your approach differs from and improves upon prior work, particularly in terms of:

* Feature engineering and selection.
* Ensemble learning strategies.
* Handling of specific attack types (e.g., replay attacks).
* Highlight any novel methodologies or insights introduced by your study.

Revise the introduction and conclusion sections to explicitly state the novel aspects of your work and its contributions to the field.
* * *
Referances

Rai, R., Grover, J., Sharma, P., and Pareek, A. (2025). Securing the CAN bus using deep learning for intrusion detection in vehicles. Scientific Reports, 15(1), 13820.

Agbo, Obinna C. "Machine Learning Based Intrusion Detection Framework for CAN Bus Vulnerabilities in Modern Vehicles." (2024).

Suwwan, R., Alkafri, S., Elsadek, L., Afifi, K., Zualkernan, I., & Aloul, F. (2021, November). Intrusion detection for CAN using deep learning techniques. In International Conference on Applied CyberSecurity (pp. 13-19). Cham: Springer International Publishing.

Devnath, Maloy Kumar. "Gcnids: Graph convolutional network-based intrusion detection system for can bus." arXiv preprint arXiv:2309.10173 (2023).

Althunayyan, Muzun, Amir Javed, and Omer Rana. "A robust multi-stage intrusion detection system for in-vehicle network security using hierarchical federated learning." Vehicular Communications 49 (2024): 100837.

Jeong, Seonghoon, et al. "X-CANIDS: Signal-aware explainable intrusion detection system for controller area network-based in-vehicle network." IEEE Transactions on Vehicular Technology 73.3 (2023): 3230-3246.

Saheed, Yakub Kayode, and Joshua Ebere Chukwuere. "Xaiensembletl-iov: A new explainable artificial intelligence ensemble transfer learning for zero-day botnet attack detection in the internet of vehicles." Results in Engineering 24 (2024): 103171.

Nwakanma, Cosmas Ifeanyi, Love Allen Chijioke Ahakonye, Judith Nkechinyere Njoku, Jacinta Chioma Odirichukwu, Stanley Adiele Okolie, Chinebuli Uzondu, Christiana Chidimma Ndubuisi Nweke, and Dong-Seong Kim. "Explainable artificial intelligence (XAI) for intrusion detection and mitigation in intelligent connected vehicles: A review." Applied Sciences 13, no. 3 (2023): 1252.

Altalbe, Ali. "Enhanced intrusion detection in in-vehicle networks using advanced feature fusion and stacking-enriched learning." IEEE Access 12 (2023): 2045-2056.

Ahmed, Usman, Zheng Jiangbin, Ahmad Almogren, Sheharyar Khan, Muhammad Tariq Sadiq, Ayman Altameem, and Ateeq Ur Rehman. "Explainable AI-based innovative hybrid ensemble model for intrusion detection." Journal of Cloud Computing 13, no. 1 (2024): 150.

Reviewer 1 ·

Basic reporting

The authors seem to selectively ignore some of the advanced intrusion detection algorithms that address CAN security, especially those based on deep learning. These methods are usually more accurate. In addition, the authors report much lower results for DCNN than those reported in the literature, which is quite unreasonable.

Experimental design

N/A

Validity of the findings

I still suspect the results and novelty of the proposed methods.

Additional comments

N/A

Reviewer 2 ·

Basic reporting

the authors have responded to my comments

Experimental design

the authors have responded to my comments

Validity of the findings

the authors have responded to my comments

Additional comments

the authors have responded to my comments

Reviewer 3 ·

Basic reporting

Thank you to the authors for the efforts in revising the manuscript. My previous comments have been addressed.

Experimental design

no comment

Validity of the findings

no comment

Additional comments

no comment

---

## Round 0.3 · Minor Revisions

Please follow the reviewers' comments.

Reviewer 1 ·

Basic reporting

N/A

Experimental design

1. Baseline methods are further required to be updated.

2. The quality of the Figures is further required to be polished and clarified.

Validity of the findings

N/A

Additional comments

The paper may undergo minor revisions and then be accepted.

Reviewer 4 ·

Basic reporting

There are some minor spelling and grammar issues on lines 327, 684,714, 728-729, and in Table 1, Figure 1, Figure 3, Table 9, see also attached PDF for clarification.

Lines 239 to 242 seem to be in the wrong place, because all the other points reference attack types, but this point seems to be a description of BEPCD.

Experimental design

Thank you for providing the code, however, it could be improved by adding a README to clarify the differences between notebooks in order to increase reproducibility.

Validity of the findings

no comment

Annotated reviews are not available for download in order to protect the identity of reviewers who chose to remain anonymous.

---

## Round 0.4 · Minor Revisions

Please fix the last comments.

Reviewer 1 ·

Basic reporting

I have no further comments on this paper. It is suitable for acceptance.

Experimental design

N/A

Validity of the findings

N/A

Additional comments

N/A

Reviewer 2 ·

Basic reporting

Accepted

Experimental design

Accepted

Validity of the findings

Accepted

Additional comments

Accepted

Reviewer 3 ·

Basic reporting

no comment

Experimental design

no comment

Validity of the findings

no comment

Additional comments

Thanks for addressing the previous comments.

Reviewer 4 ·

Basic reporting

Spelling mistake on line 323 and line 13 in Algorithm 1

Line 610 says CICIOV 2014, instead of CICIOV 2024

On lines 377-379 you say that both Stacking and Blending integrates RF, ET and XGBoost while in Figure 3, Blending also includes CatBoost. Furthermore, in Algoritm 1, Stacking uses CatBoost, so which is correct? In the code, you use RF, CatBoost and XGBoost for Blending and RF, ET and XGBoost for Stacking.

Experimental design

no comment

Validity of the findings

no comment

---

## Round 0.5 · accepted · Accept

Thank you for contributing to PeerJ.